# Multiphysical description of atmospheric pressure interface chemical ionisation in MION2 and Eisele type inlets

Henning Finkenzeller[1,2], Jyri Mikkilä[2], Cecilia Righi[1], Paxton Juuti[2], Mikko Sipilä[1], Matti Rissanen[3,4], Douglas Worsnop[1,5], Aleksei Shcherbinin[2], Nina Sarnela[1], and Juha Kangasluoma[1,2]

[1]Institute for Atmospheric and Earth System Research / Department of Physics, Faculty of Science, University of Helsinki, Helsinki, FI-00560, Finland
[2]Karsa Ltd., FI-00560, Finland
[3]Aerosol Physics Laboratory, Physics Unit, Faculty of Engineering and Natural Sciences, Tampere University, Tampere, FI-33720, Finland
[4]Department of Chemistry, University of Helsinki, Helsinki, FI-00014, Finland
[5]Aerodyne Research Inc., Billerica, Massachusetts 01821, USA

**Correspondence:** Henning Finkenzeller (henning.finkenzeller@helsinki.fi)

**Abstract.** Chemical ionisation inlets are fundamental instrument components in chemical ionisation mass spectrometry (CIMS). However, the sample gas and reagent ion trajectories are often understood only in a general and qualitative manner. Here we evaluate two atmospheric pressure chemical ionisation inlets (MION2 and Eisele type inlet) with computational fluid dynamics 3D physico-chemical models regarding the reagent ion and sample gas trajectories and estimate their efficiencies of reagent

ion production, reagent ion delivery from the ion source volume into the ion–molecule mixing region, and the interaction between reagent ions and target molecules. The models are validated by laboratory measurements and quantitatively reproduce observed sensitivities to tuning parameters, including ion currents and changes in mass spectra. The study elucidates how the different transport and chemical reactions proceed within the studied inlets, that space charge can already be relevant at ion concentrations of as low as $10^7\,\mathrm{cm}^{-3}$, and compares the two investigated inlet models. The models provide insights into how

to operate the inlets and will help in the development of future inlets that further enhance the capability of CIMS.

## 1   Introduction

Mass spectrometry (MS) requires target molecules to be electrically charged to determine their mass-to-charge ratios and infer their atomic composition. Electrically neutral sample molecules require ionisation prior to MS. In chemical ionisation (CI) inlets, gas-phase ions created outside the sample inlet are introduced into the ion-molecule mixing region (IMR), where they

interact with the sample gases by ion attachment, scattering or recombination (Passananti et al., 2019). The ions are transmitted from the IMR to a mass spectrometer at lower pressure through a suitable aperture.

For accurate and precise mass-spectrometric measurements, chemical ionisation inlets should introduce reagent ions into the IMR at sufficient concentration, enable a suitable reaction time in the IMR, avoid transport losses within the inlet, and minimise other measurement biases, such as contamination. Higher reagent ion concentrations not only allow better detection

limits, but also delay reagent ion depletion in conditions with high analyte concentrations, thereby enhancing the dynamic range. Additionally, inlets should be robust and easy to operate.

A variety of CI inlet designs have been developed since the emergence of CIMS, which differ in used reagent ions, supply and generation of the reagent ions (radioactive source, x-ray, VUV lamp, corona discharge, electrospray), reaction time, and IMR pressures. Reduced pressure inlets (IMR at fractions of atmospheric pressure) are used to suppress multiple collisions between the reagent ion and target molecules (required for e.g. proton-transfer-reaction mass spectrometry (PTR-MS, Yuan et al. (2017))) and the formation of reagent ion oligomers. Atmospheric pressure interface (API) CI inlets minimise the introduction of turbulence in the sample flows and reach excellent detection limits due to high reagent ion and sample gas concentrations in the IMR. Prominent examples of API CI inlets used in atmospheric science are the Eisele type (Eisele and Tanner, 1991, 1993; Tanner and Eisele, 1995; Tanner et al., 1997; Mauldin III et al., 1998; Sipilä et al., 2015) and MION inlet (Rissanen et al., 2019; Wang et al., 2021; Huang et al., 2021; Rissanen, 2021; Iyer et al., 2021; Shen et al., 2022; Finkenzeller et al., 2023; He et al., 2023; Partovi et al., 2023).

The description of the inlets in the literature is currently limited to schematics or conceptual modelling (Sipilä et al., 2015). While tuning the control parameters of inlets (voltages and gas flow rates) is generally required before measurements, the influence of individual control parameters on the detailed processes within the instruments may not always be straightforward and evident to the users. Examples of well-known but little-understood features are the formation of $Br_3^-$ or $I_3^-$ in $Br^-$ or $I^-$ CIMS, the presence of reagent precursor gas in the IMR of the Eisele inlet, or the sensitivity of the Eisele inlet to voltages and the exact insertion depth of the sample tube. Using physico-chemical modelling together with measurements, this study aims to provide a clear picture of the respective gas and ion trajectories within the inlets, to compare the two inlet designs, and to identify limitations and avenues for the development of improved inlet designs with higher reagent ion yields and other ion-chemistries.

## 2 Methods

### 2.1 Physico-chemical processes within the inlets and representation in model

The mechanisms influencing the trajectories and distribution of the reagent ions and target molecules are: (1) the ion generation from a source gas; (2) the gas flow throughout the inlet, i.e., sample and auxiliary gas flows; (3) electric fields from ion optics and ions themselves; (4) diffusion; (5) the transport of molecules and ions in the electro-convective field; (6) the chemical conversion between species in the gas phase; and (7) interaction (loss or conversion) of gas-phase species on surfaces. These processes are quantitatively and spatially represented in a stationary model implemented in COMSOL Multiphysics 6.1. The controlling factors and metrics along the ion trajectories are described in the following.

### 2.1.1 Ion generation in source region

In MION2 and Eisele type inlets, precursor gases are split into positive and negative ions by introducing energy from radioactive sources or x-ray lamps (Anttalainen et al., 2021). The detailed mechanism of ion formation from precursor gases is not trivial – e.g., ionisation radiation may initially ionise the bath gas, not the precursor molecules itself – and is not the focus of this study. Here, the primary production of ions is approximated as splitting up a precursor gas $R^cR$ into the reagent ion $R^{\pm}$ and a counter ion $R^{c\mp}$ of opposite polarity at a prescribed rate.

Ion–ion recombination is extremely fast, with typical bimolecular rate coefficients of $1.7 \cdot 10^{-6}\,\mathrm{cm^3\,s^{-1}}$ (Zauner-Wieczorek et al., 2022), four orders of magnitude faster than radical–radical recombination. As a binary process, it becomes progressively more important at higher ion concentrations. The lifetime of a population against recombination at $[R^{\pm}] = [R^{c\mp}] = 10^9\,\mathrm{cm^{-3}}$ is only $\sim 1\,\mathrm{ms}$. In absence of efficient charge separation, ion production is near-completely offset by recombination, and a doubling of the initial ion production rate $P$ does not double the established concentrations in the ionisation volume but rather increases them by 40 %. The obvious method to separate newly-generated ions of opposite polarity is the application of electric fields. A metric assessing the capability of an inlet to generate reagent ions is the reagent ion concentration $c_S$ downstream of the ion source volume, as it presents the upper limit for the attainable ion concentrations in the IMR.

### 2.1.2 Electro-convective ion transport to IMR

The ion bulk velocity $\boldsymbol{v}_{\mathrm{ion}}$ is the sum of advective and electrophoretic velocity, i.e., the advective flow field and the electric field are additive:

$$\boldsymbol{v}_{\mathrm{ion}} = \boldsymbol{v}_{\mathrm{conv.}} + \mu\boldsymbol{E} \tag{1}$$

Here, $\boldsymbol{v}_{\mathrm{ion}}$ is the flow velocity $[\mathrm{cm\,s^{-1}}]$, $\mu$ $[\mathrm{cm\,s^{-1}\,cm\,V^{-1}}]$ the electrical mobility, and $\boldsymbol{E}$ $[\mathrm{V\,cm^{-1}}]$ the electric field strength. The ion transport may occur predominately advectively, convectively (i.e., by advection and diffusion), or electrophoretically in different parts of the inlet.

Externally constrained advective and electrophoretic fields (i.e., disregarding space charge induced fields) are divergence free, i.e., the only field sources and sinks are the boundary conditions (gas inlet and outlet, electrodes); the in-flux $I$ into a given volume is equal to its out-flux. If for a given $I$ the out-flux area $A$ changes, the out-flow velocity $v$ changes inversely:

$$\frac{I}{A} = const. \tag{2}$$

Here, $A$ is the flow cross section area $[\mathrm{m^2}]$. Therefore, ion mixing ratios are conserved along electro-advective streamlines, and ion concentrations $c$ are conserved if the gas density does not change. Analogous to narrowing riverbanks that increase the water flow velocity ($v$) by reducing the cross-section area of the flow ($A$) but do not change the composition of the water ($c$), electric fields defined by electrodes affect the ion trajectories without changing their concentration. In absence of collisional focusing (e.g., Kelly et al., 2010), ion concentrations along streamlines do not increase.

The reagent ion concentration at the pinhole $c_P$, the orifice connecting the IMR and the mass spectrometer, may reach but cannot exceed the upstream ion concentration at the ion source $c_S$. The theoretical maximum ion current at the pinhole $I_{max}$

is hence the product of the ion concentration at the ion source region $c_S$ and the flow rate at the pinhole $J_P$:

$$I_{max} = J_P c_S \tag{3}$$

To achieve a high concentration of reagent ion in the IMR hence requires to (1) create a high initial concentration of ions, and to (2) efficiently deliver the ions to the IMR.

Efficient ion transport means that the initially generated ion concentration is maintained along streamlines to the IMR. The ion delivery efficiency $\eta_D$ is defined here as

$$\eta_D = \frac{c_P}{c_S} \tag{4}$$

Here, $c_P$ is the ion concentration at the pinhole (the aperture to the mass spectrometer), $c_S$ is the ion concentration at the ion source.

The space charge of ions distributed in space needs to be considered if ion concentrations are so high that the induced electric fields are comparable in magnitude to the prescribed electro-advective field. Gauss's law describes the creation of electric fields due to charge distributions:

$$\nabla \cdot \boldsymbol{E} = \frac{\rho}{\varepsilon} \tag{5}$$

Here, $\varepsilon$ [F m$^{-1}$] is the permittivity, $\rho = ce$ [C cm$^{-3}$] the charge concentration. For a beam of singly charged ions with concentration $c = 10^7$ cm$^{-3}$, the space charge-induced electric field 5 mm off the beam centre axis is $E = 9$ V cm$^{-1}$, corresponding to a radial drift velocity $v = 22$ cm s$^{-1}$ ($\mu(\mathrm{NO_3^- - N_2}) = 2.4$ cm$^2$ V$^{-1}$ s$^{-1}$). This is significant when compared to typical advective velocities of 1 m s$^{-1}$. Space charge matters even at these relatively low concentrations.

Diffusion of the ions perpendicular to the electro-advective streamlines needs to be considered wherever concentration gradients are significant, especially at the edges of ion beams.

The electro-convective streamlines that connect the ionisation volume and the pinhole define what part of the ionisation volume actually contributes to the delivery of ions into the pinhole. As the product of flow velocity and area is constant for a given flow rate(eq. 3), the increase of $E$ necessarily requires the area of usable extraction $A$ to become smaller. Faster transport minimises space-charge and diffusional losses, but generally decreases the source concentration $c_S$. This is analogous to dumping a compound into a river at a constant rate: the faster the river flows, the lower the resulting compound concentration.

### 2.1.3 Reaction time in ion–molecule mixing region

The reaction time $t$ between analyte molecules A and reagent ions $R^\pm$ influences the abundance of analyte–reagent ion clusters, in addition to the concentrations of analyte and reagent ions, and the clustering reaction rate constant $k$. The model allows to elegantly determine $t$. Consider a bi-molecular clustering reaction $A + R^\pm \to AR^\pm$ with reaction rate coefficient $k$. If the reactants A and R are not significantly consumed in the reaction, then the concentrations [A] and [R] at a given time are representative for the entire reaction time and the cluster concentration at the pinhole is a simple function of $t$:

$$[AR^\pm] = tk[A][R^\pm] \tag{6}$$

The concentrations and reaction times along different trajectories to the pinhole are generally not the same. The average reaction time $t_{\mathrm{avg}}$ that considers different trajectories to the pinhole with different reaction times and concentrations is given by the integral pinhole currents $I$ for a given pinhole flow $J_P$:

$$t_{\mathrm{avg}} = \frac{I_{AR\pm} J_P}{k I_A I_{R\pm}} \tag{7}$$

### 2.1.4 Theoretical calibration factor and detection limit

The calibration factor $C_A$ $[\mathrm{cm}^{-3}\,\mathrm{cps}^{-1}\,\mathrm{cps}]$ for a target compound is a result of $k$ and $t_{\mathrm{avg}}$:

$$C_A = c_A \frac{I_{R\pm}}{I_{AR\pm}} = \frac{I_A}{J_P} \frac{I_{R\pm}}{I_{AR\pm}} \tag{8}$$

Here, $c_A$ is the concentration of compound A. $C_A$ depends on the compound- and detector-specific detection sensitivities and needs to be determined experimentally. The detection limit $\Lambda$ additionally depends on the magnitude of $I_{R\pm}$ and the $I_{AR\pm}$ baseline.

## 2.2 Model setup

The inlet geometries are approximated by meshes consisting of multiple million volumes, including surface layers. The symmetry of the inlets is exploited to limit the modelling to half (MION2) or even a quarter (Eisele) of the full geometry. The convective flow field is determined prior to determining the concentrations of chemical compounds in the electro-convective field. This reduces the complexity of the numerical system and is justified as the convective field is not influenced by the transport of dilute molecules. The model uses a temperature of $293\,\mathrm{K}$ and gas reference pressure of $1\,\mathrm{atm}$. The model assumes laminar flow and uses prescribed rates for the exhaust, pinhole, and auxiliary flows as constraints. For the modelling of the Eisele-type inlet, $10\,\mathrm{slpm}$ sample, $20\,\mathrm{slpm}$ sheath, and $1\,\mathrm{slpm}$ flow to the mass spectrometer are used (Tanner and Eisele, 1995). For the MION2 inlet, $20\,\mathrm{slpm}$ exhaust flow (Wang et al., 2021) and $0.8\,\mathrm{slpm}$ flow to the mass spectrometer are used. The auxiliary reagent, purge, and reagent exhaust flow are $J_R = 10\,\mathrm{smlpm}$, $J_{RE} = 50\,\mathrm{smlpm}$, and $J_{RP} = 100\,\mathrm{smlpm}$.

The electric fields are constrained by specified electrode potentials and the space charge of the ions. The model uses the electric mobility constant $\mu = 2.4\,\mathrm{cm^2\,V^{-1}\,s^{-1}}$ ($\mu(\mathrm{NO_3{-}air}, T = 24\,^\circ\mathrm{C})$, Xuemeng Chen, personal communication (compare Steiner et al., 2014)) for all ions, equivalent to a diffusivity constant $D = 0.062\,\mathrm{cm^2\,s^{-1}}$. The mobility constant determines what electric field strength magnitude is required. The variability in electrical mobility between different ions is small and not critical in this study (Hwang et al., 1989; Hwang and Su, 1990; De Andrade et al., 1992; Filippov et al., 2017; Cussler, 2009), but could be significant in systems with light reagent ions and large clusters, i.e., proton transfer reaction mass spectrometers.

In this study, simplified chemistry schemes for the operation with either $\mathrm{NO_3^-}$ or $\mathrm{Br^-}$ as reagent ion were used. The reagent ions $\mathrm{NO_3^-}$ and $\mathrm{Br^-}$ are produced from splitting the source gas (nitric acid, $\mathrm{HNO_3 \rightarrow H^+ + NO_3^-}$, or bromomethane, $\mathrm{CH_3Br \rightarrow CH_3^+ + Br^-}$). $\mathrm{Br^-}$ is in practice often generated from dibromomethane, which could in principle donate two $\mathrm{Br^-}$ and has different ionisation properties; in this study, dibromomethane can be considered interchangeable with bromomethane if only a single dissociation is assumed to take place. As proxy for target molecules, dilute sulphuric acid $\mathrm{H_2SO_4}$ is modelled to be

contained in the sample flow at a mixing ratio of $1\,\mathrm{ppt}$. It reacts kinetically with $\mathrm{Br}^-$ and $\mathrm{NO_3^-}$ to form $\mathrm{H_2SO_4 \cdot Br^-}$ and $\mathrm{H_2SO_4 \cdot NO_3^-}$. The magnitude of the $\mathrm{H_2SO_4}$ abundance is not critical for the interpretation of the modelling results as long as the clustering with the reagent ion does not substantially reduce the reagent ion concentration. While the precursor gases are assumed to be in steady state with the surfaces, $\mathrm{H_2SO_4}$ and all ions are assumed to be lost efficiently to the inlet surfaces. $\mathrm{Br}^-$, $\mathrm{Br_3^-}$, and $\mathrm{H_2SO_4 \cdot Br^-}$ surface-uptake is assumed to lead to the re-emission of $\mathrm{Br_2}$ (at respective stoichiometric ratios). The same chemistry is used in both inlets, facilitating a direct comparison.

## 2.3  Laboratory measurements

Measured electrode currents due to the absorption of attracted ions were used to constrain the production rate of ion pairs in the ionisation volume. The currents to the two topmost electrodes of the MION2 inlet ($4 \cdot 10^{-11}\,\mathrm{A}$, attraction of $\mathrm{H}^+$) and for the ion cage of the Eisele inlet ($6 \cdot 10^{-11}\,\mathrm{A}$, attraction of $\mathrm{H}^+$, negligible adsorption of $\mathrm{NO_3^-}$) were determined via the voltage-drop across the internal $10.\,\mathrm{M\Omega}$ resistor dedicated to measure voltages in a simple multimeter (Tenma 72-2595). The voltage-drop of $0.6\,\mathrm{mV}$ is measurable by the voltmeter and does not constitute a measurement bias under the test conditions. For both inlets, the model reproduced the measured currents assuming a production rate $P = 6 \cdot 10^7\,\mathrm{cm^3\,s^{-1}}$, equivalent to an ion–ion recombination determined steady state concentration of $6 \cdot 10^6\,\mathrm{cm^{-3}}$. Bromide spectra with the MION2 inlet were measured with a long time of flight MS (LTOF, Tofwerk AG, Switzerland); the voltage-dependent ion current to the mass spectrometer in the Eisele setup was measured with a high resolution time of flight MS (HTOF, Tofwerk AG, Switzerland). The Eisele inlet was used with a single x-ray source (Hamamatsu L12535). $\mathrm{H_2SO_4}$ or other target gases were not employed in the laboratory experiments but treated in the modelling only.

## 3  Results

### 3.1  MION2 inlet

Figure 1 shows the geometry and physical quantities in the MION2 inlet, using nitric acid as reagent gas and sulfuric acid as sample gas. Sample gas is drawn into the IMR (inner diameter $22\,\mathrm{mm}$, length from source centre to orifice $33\,\mathrm{mm}$) and to the pinhole and exhaust advectively, while auxiliary flows in the ion source region are minimal (Fig. 1b). Assuming an interface upstream of the MION2 inlet that creates a fully developed laminar flow (Reynolds number $Re \approx 2100$, using $D = 20\,\mathrm{mm}$, $u = 1.6\,\mathrm{m\,s^{-1}}$, $\nu = 1.48 \cdot 10^{-5}\,\mathrm{m^2\,s}$), the flow velocity profile is parabolic throughout the sample tube and IMR close up to the pinhole plate, where the flow splits to the exhaust and pinhole. In the ion source, predominately electric fields, generated by 20 electrodes, transport the ions (Fig. 1c). Figure 1d shows the electro-advective field and streamlines for ions. Note that the transfer from electric to convective transport occurs where the electric and convective streamlines are approximately perpendicular to each other. Sample gas (here, $\mathrm{H_2SO_4}$ and $\mathrm{H_2O}$, Fig. 1e and f) is kept out of the ion source volume by a small purge flow $J_{RP}$, and reagent gas provided in flow $J_R$ is likewise contained to the ionisation volume only (Fig. 1g) with a small exhaust flow $J_{RE}$. The ionisation of the precursor gas $\mathrm{HNO_3}$ leads to formation of the complementary ions $\mathrm{H}^+$ and $\mathrm{NO_3^-}$

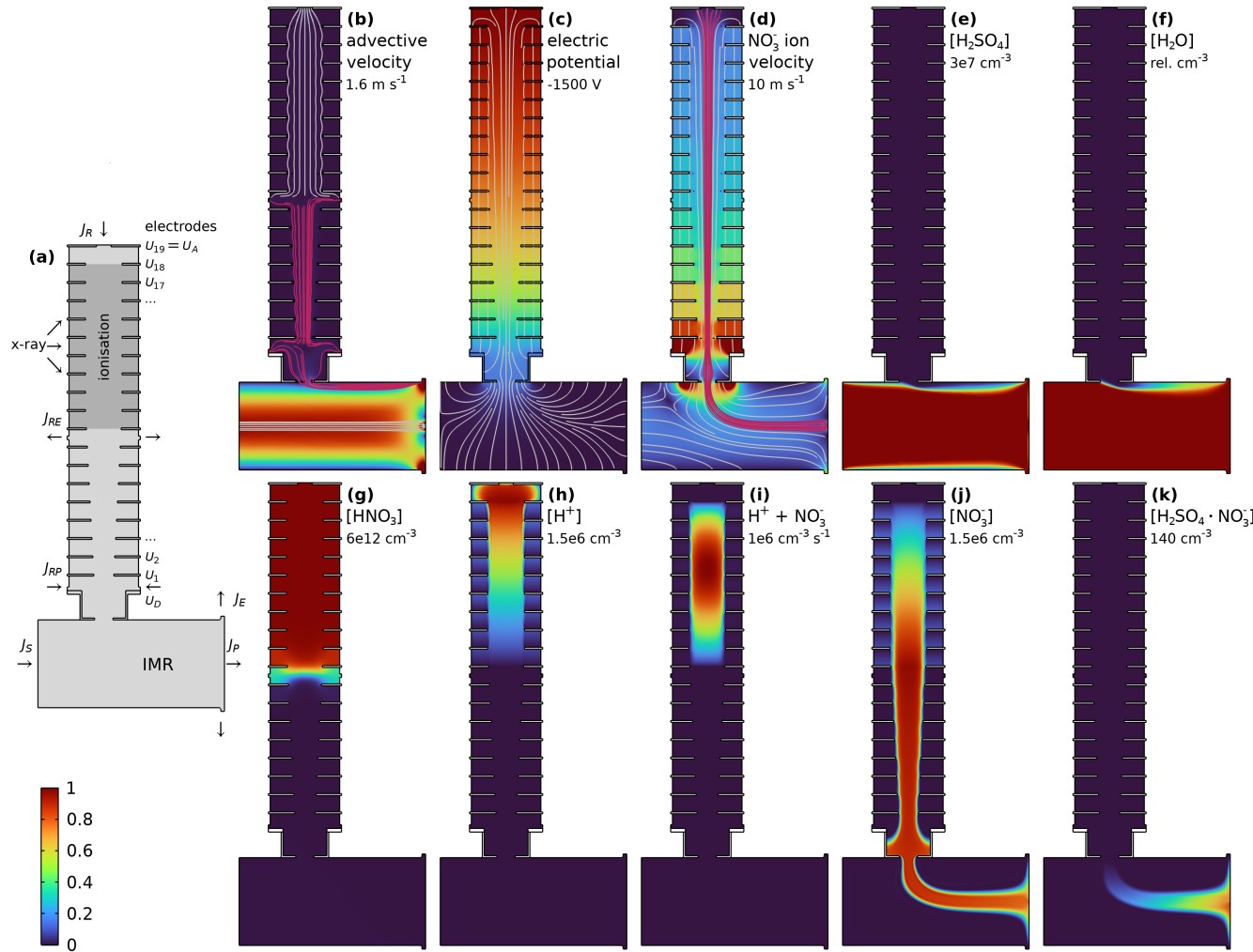

**Figure 1.** Modelled physical quantities in the MION2 inlet centre plane using $NO_3^-$ as the reagent ion. The colour scale ranges from 0 to the maximum described in each panel. Panel d shows the electro-advective velocity for anions with $\mu = 2.4\,\text{cm}^2\,\text{V}^{-1}\,\text{s}^{-1}$. Used settings: Accelerator voltage $U_A = -1500\,\text{V}$, deflector voltage $U_D = -110\,\text{V}$, exhaust flow $J_E = 20\,\text{slpm}$, pinhole flow $J_P = 800\,\text{sccm}$, reagent gas flow $J_R = 15\,\text{sccm}$, reagent exhaust flow $J_{RE} = 50\,\text{sccm}$, reagent purge flow $J_{RP} = 100\,\text{sccm}$.

(Fig. 1h and j), which are separated by electric fields. Ion–ion recombination is strongest in the centre of the ionisation volume
(Fig. 1i). $NO_3^-$ is transported along the ion-flow streamlines into the IMR. The clustering of $NO_3^-$ and $H_2SO_4$ leads to buildup of $H_2SO_4 \cdot NO_3^-$ clusters (Fig. 1k).

Figure 2 shows the simplified bromine chemistry in the MION2 inlet when bromomethane is used as the reagent gas. Here, bromomethane (Fig. 2a) is split into methylium and bromide (Fig. 2b and d). $Br_2$ concentrations result from the absorption of

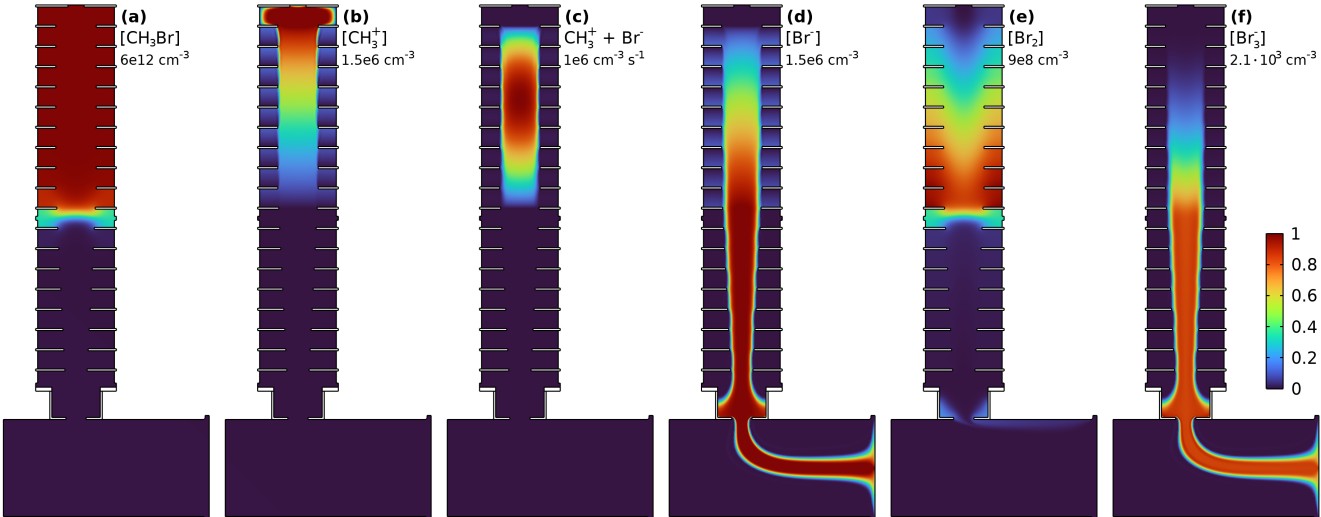

**Figure 2.** Bromine chemistry in the MION2 inlet centre plane. $Br_2$ forms in the wall uptake of $Br^-$ and $Br_3^-$. $Br_3^-$ forms from recombining $Br^-$ and $Br_2$. The colour scale ranges from 0 to the maximum described in each panel.

$Br^-$ (and $Br_3^-$) to the surfaces and re-emission as $Br_2$ (Fig. 2e). $Br_3^-$ is formed in the model by the kinetic recombination of $Br^-$ and $Br_2$ (Fig. 2f).

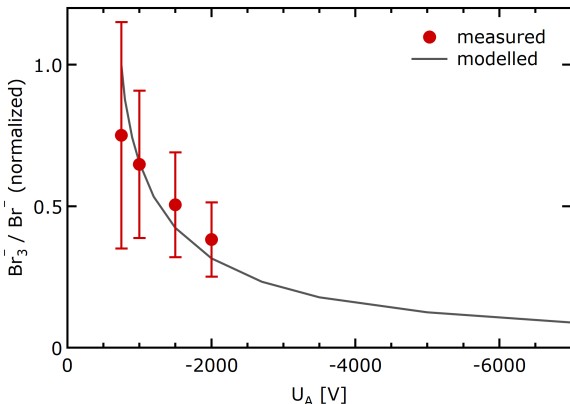

**Figure 3.** $Br_3^-$ sensitivity in measurements and model. At low voltages, the longer residence time in the ion source leads to enhanced relative formation of $Br_3^-$. Error bars indicate the measurement standard deviation.

Figure 3 shows the voltage dependent ratio of $Br_3^-$ and $Br^-$ delivery in MION2 in measurements and the model. The model is able to reproduce the observed trend, which is due to the voltage-dependent reaction time within the $Br_2$-filled volume: At low electro-convective velocities there is more time for the $Br_2$–$Br^-$ clustering to occur. The model-predicted relative $Br_3^-$ abundance is on the order of permil for the studied conditions. Higher $Br_3^-$ abundances would establish if either higher $Br^-$

concentrations led to stronger $Br_2$ production or dilution in the ionisation volume was reduced by a slower supply of reagent gas. Additionally, the $Br_2$–$Br^-$ recombination, a neutral–ion clustering, likely occurs at a rate faster than the neutral–neutral collision rate currently used in the model. As the concentrations of neither $Br_3^-$ nor $Br^-$ were measured quantitatively in the mass spectrometer, due to compound-specific transmission and detection efficiencies, only scaled ratios are shown in Fig. 3. The $Br_3^-$ formation mechanism could be applied to analogously explain the formation of $I_3^-$ in iodide CIMS.

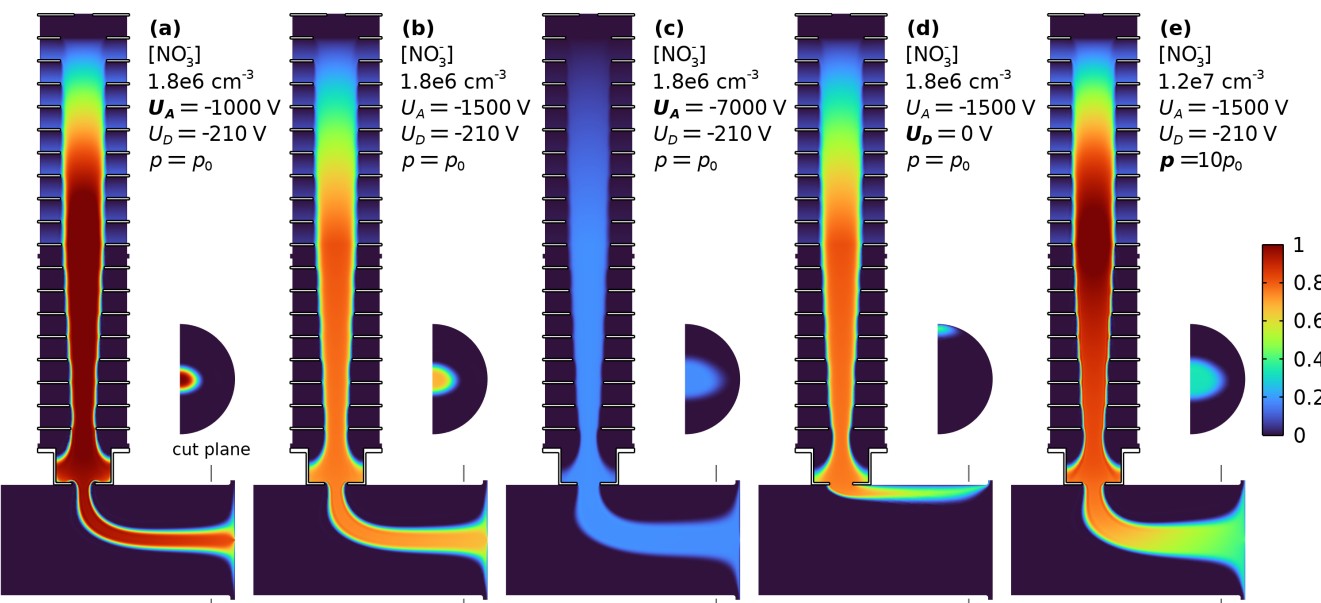

**Figure 4.** Sensitivities of $NO_3^-$ concentrations in MION2 inlet to different acceleration voltages $U_A$ (a–c), deflector voltage $U_D = 0\,V$ for deactivation (d), and primary ion production rate (e). The semi circle areas show the ion concentration in the cut plane $5\,mm$ in front of the orifice. The colour scale ranges from 0 to the maximum described in each panel. Figures a–d use the same colour scale. The width of the ion beam increases for larger voltages, while the extracted concentrations slightly decrease. At concentrations of $10^7\,cm^{-3}$ space charge leads to a spreading of the ion beam, the concentration at the pinhole is lower than at the ionisation volume.

Figure 4 shows how the accelerator voltage $U_A$ and deflector voltage $U_D$ affect the ion trajectories and concentration. The accelerator voltage $U_A$ influences the concentration of ions extracted from the ionisation volume and controls the width of the reagent ion beam injected into the sample tube and IMR (Figure 4a–c). The decrease of source concentration $c_S$ for higher $|U_A|$ is expected as for faster electro-convective transport the ions, produced at a constant rate, are distributed over a wider volume. To maximise the ion delivery to the pinhole, all gas drawn into the pinhole should be illuminated by ions, i.e., the width of

the ion beam should be as wide as the pinhole flow collection aperture. Beams slightly larger than geometrically needed help to counter diffusion losses and reduce the sensitivity to choosing the deflector voltage $U_D$ correctly. The deflector voltage $U_D$ controls how deep the ions are pushed radially into the convective sample tube. If $U_D$ is too repellent, ions are pushed to the opposite side of the sample tube. If it is insufficiently repellent, the reagent ions do not penetrate into the pinhole flow. If chosen correctly, the electro-advective streamlines connect the pinhole and the ionisation volume (Fig. 1d), and the distribution of ions

in the IMR close up to the pinhole is in good approximation rotationally symmetric (Fig. 4). The marginal beam compression in the ion injection direction is due to the advective velocity being largest in the plane of injection. Figure 4d shows that when the deflector voltage $U_D$ is set to ground potential, i.e., minimal repulsion, some ions are still injected. To fully suppress ion injection, the x-ray source should be switched off. Figure 4e illustrates how space charge progressively matters at higher ion concentrations. Here, at $[\mathrm{NO_3^-}] = 1.2 \cdot 10^7\,\mathrm{cm^{-3}}$, space charge leads to a widening of the ion beam, and the ion concentration notably decreases from the source to the pinhole.

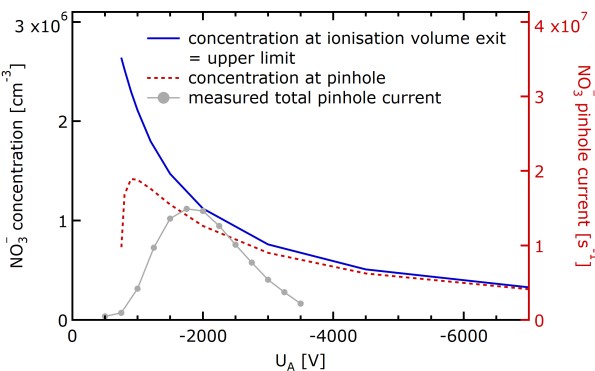

**Figure 5.** Ion concentration conservation from the MION2 ionisation volume to the pinhole, as function of the accelerator voltage $U_A$. The red dashed trace shows the ion concentration respectively the current entering the pinhole. The concentration and current axis are connected by the pinhole flow rate $J_P$.

Figure 5 shows for the MION2 inlet the $\mathrm{NO_3^-}$ ion source concentration $c_S$, i.e., the concentration of reagent ions just downstream of the ionisation volume, for different accelerating voltages $U_A$. While stronger electric fields lead to higher fluxes, they do not increase ion concentrations (eq. 2). The extracted ion concentration $c_S$ are in the low $10^6\,\mathrm{cm^{-3}}$ range. Figure 5 also shows in red the ion current at the MION2 pinhole for different accelerating voltages, respectively the ion concentration at the pinhole. The factor connecting the source concentration scale and the ion current scale is the flow rate $J_P$. At $|U_A| < 750\,\mathrm{V}$, the deflector voltage $U_D = -210\,\mathrm{V}$ creates an electric barrier, and reagent ions do not enter from the ionisation source into the IMR. The closure between measured and modelled pinhole currents is only qualitative. The onset of transmission occurs at a similar voltage, but it is less sharp in measurements. The maximum pinhole current is measured at a higher $|U_A|$ than predicted. Although the deflector voltage $U_D$ was chosen in the measurement to maximise the ion delivery, it is possible that the ion beam was not always axially centred to be contained within or to fully illuminate the pinhole flow. In very narrow ion beams (low $|U_A|$, compare Fig. 4a) that are not aligned with the flow going to the pinhole, most ions entering the IMR would be lost to the exhaust flow ($J_E$). This is a plausible explanation for the gradual onset of the observed measured total pinhole current.

We note that the measured pinhole current in Figure 5 apparently decreases faster towards higher absolute accelerator voltages $|U_A|$ than expected from model simulations. Insufficient centring of the beam does not explain this observation, as the ion concentration within the beam varies only slightly. While shying from pinpointing a specific mechanism, we hypothesise

that the effect originates in the volume controlled by $U_A$, i.e., the ionisation volume or the buffer volume between the ionisation volume and IMR. Actual ion mobilities considerably higher than assumed in the model at high $|U_A|$ would lead to a more rapid decrease. A lower degree of reagent ion hydration and cluster formation between the reagent ion and reagent gas at high field strength would increase the effective electrical mobility. The field strength sensitivity of the reagent ion mobility itself is less likely to be significant because of the still relatively weak field strength (Viehland and Mason, 1995). Space charge losses during transport (especially within the IMR) are found to be not yet significant for the ion concentrations of few $10^6\,\mathrm{cm}^{-3}$. In the model, the ion delivery efficiency $\eta_D$ (eq. 4) is larger than 90 %, essentially unity, for $|U_A| > 3000\,\mathrm{V}$: The ion concentration is maintained from the ion source to the pinhole.

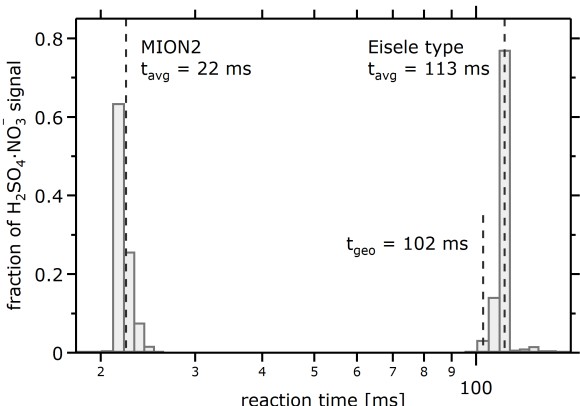

**Figure 6.** Model-derived histogram of the reaction times between reagent ion and analyte for the MION2 and Eisele type inlet. Different trajectories exhibit reaction times differing by several percent. $t_{geo}$ is the geometrical interaction, derived from the centre flow velocity and the length of the IMR.

The reaction time $t$ for the standard setup is $22\,\mathrm{ms}$ (eq. 7, Fig. 6). This is even shorter than reported values of $30\,\mathrm{ms}$ (Rissanen et al., 2019) or $35\,\mathrm{ms}$ (He et al., 2023) in the literature, which, however, is not specific about how these values were determined.

### 3.2 Eisele type inlet

Figure 7 shows the geometry (Fig. 7a) and the physical quantities in the Eisele type inlet for voltages that maximise the current to the IMR and pinhole. [1] The coordinated flow rates for sample and sheath gas minimise shear and turbulence in the IMR (inner diameter $44\,\mathrm{mm}$, length $152\,\mathrm{mm}$, Fig. 7b). The initial sample flow velocity profile is assumed to be fully developed, assuming an appropriate interface upstream of the inlet. The sheath flow profile, initialised likewise as fully developed laminar flow, quickly adjusts to the concentric tubing geometry. Figure 8 shows the velocity profile of the advective velocity upstream of the flow merging at the x-ray lamp plane and downstream after mixing at the IMR mid plane. Before merging, the flow profiles within the different channels are near-parabolic, as expected for developed laminar flows. After merging, the individual flow

---

[1]It is possible to apply a non-ground potential to the pinhole plate. The downstream voltages of the mass spectrometer need to be adjusted accordingly. Obeying volume flow conservation, putting the pinhole plate to another potential does not lead to larger currents.

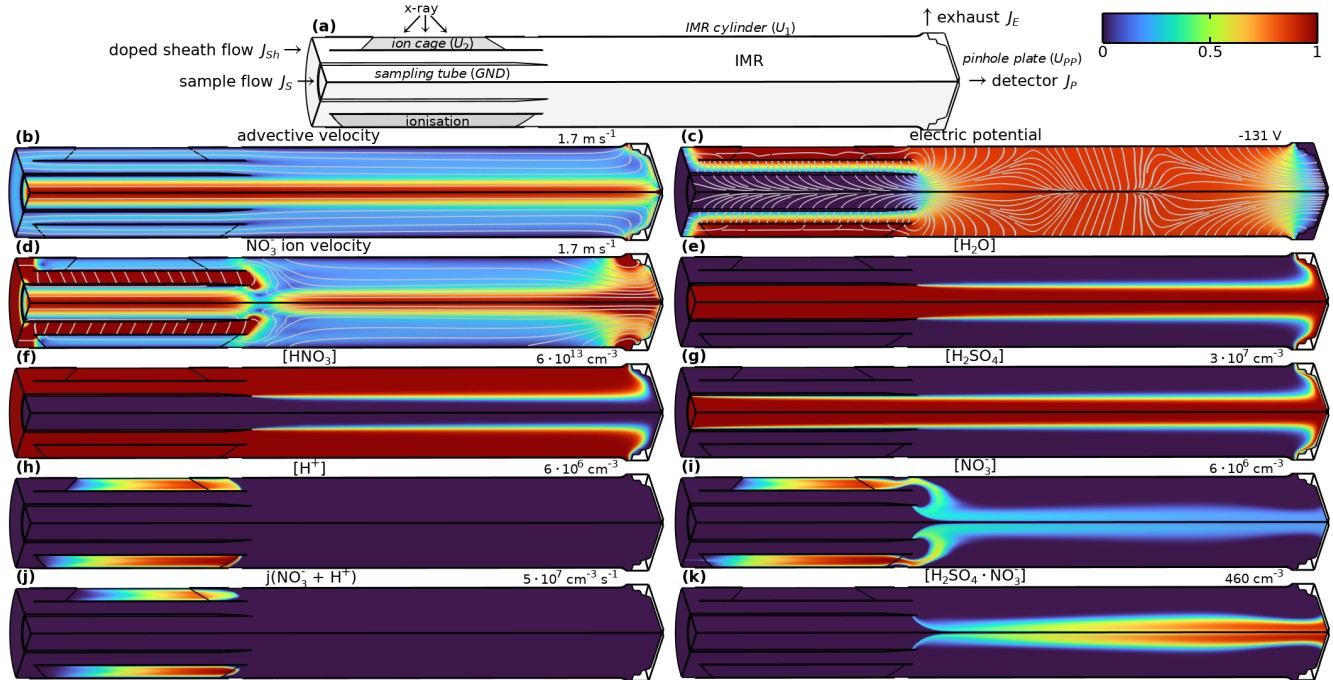

**Figure 7.** Modelled physical quantities in Eisele type inlet. Only a quarter cut is shown. The colour scale ranges from 0 to the maximum described in each panel. Panel d shows the electro-advective velocity for anions with $\mu = 2.4\,\mathrm{cm^2\,V^{-1}\,s^{-1}}$. Used settings: $U_1 = -110.7\,\mathrm{V}$, $U_2 = -98.9\,\mathrm{V}$, $U_{PP} = 0\,\mathrm{V}$, sheath flow $20\,\mathrm{slpm}$, sample flow $10\,\mathrm{slpm}$.

profiles combine to form a transition composite that maintains a near-parabolic shape in the innermost $5\,\mathrm{mm}$ with a pronounced maximum at the centre line and a rather flat shoulder with low velocity. The profile is the result of the relatively little interaction with the IMR surface after merging: The IMR radius ($22\,\mathrm{mm}$) is relatively large in comparison to the total IMR length ($15\,\mathrm{cm}$). Additionally, the velocity profile at the downstream end of the IMR (close to the pinhole plate) is not parabolic, either. The Reynolds number $Re \approx 1840$ (using $D = 16\,\mathrm{mm}$, $u = 1.7\,\mathrm{m\,s^{-1}}$, $\nu = 1.48 \cdot 10^{-5}\,\mathrm{m^2\,s}$) at the downstream end of the sample tube, the location most prone to cause turbulence, supports assuming laminar flow in the modelling.

The voltages $U_1$ and $U_2$ supplied in the Eisele-type inlet lead to an electric field (Fig. 7c) which is perpendicular to the convective field directly downstream of the ion source, but opposing the flow field closer to the centre line. Figure 7d shows the electro-convective field and streamlines for nitrate ions or other ions with a comparable mobility coefficient. At the exit of the sampling tube (inner diameter $22\,\mathrm{mm}$), the ion flow velocity decreases, as electrophoresis counteracts the convective flow. The reagent gas $HNO_3$ is mixed uniformly into the sheath flow (Fig. 7f). $HNO_3$ ionisation leads to the formation of $H^+$ (Fig. 7h) and $NO_3^-$ (Fig. 7i). $H^+$ and $NO_3^-$ are initially convectively transported out of the ionization volume, while recombination occurs (Fig. 7j). $H^+$ is lost to the ion cage, the least repulsive surface. $NO_3^-$ is first attracted towards the IMR cylinder. Once it has cleared the ion cage, it comes under the influence of the attractive electric field generated by the (electrically grounded) sampling tube . If the electric field is well matched to the convective field, $NO_3^-$ is transported towards the centre and then

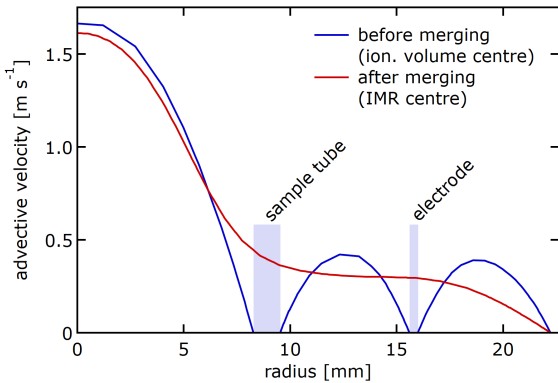

**Figure 8.** Modelled velocity profile within the Eisele-type inlet before and after merging sample and sheath flow, using $10\,\mathrm{slpm}$ sample flow and $20\,\mathrm{slpm}$ sheath flow. The composite profile establishing in the IMR has a pronounced maximum in the centre and a rather flat shoulder.

convectively to the pinhole, without significant losses of $NO_3^-$ to the surfaces. The electric gradient between the IMR cylinder and the pinhole plate leads to a focusing of the ions before entering the pinhole. The clustering of $NO_3^-$ and $H_2SO_4$ (Fig. 7g), a proxy for target species, leads to according buildup of $H_2SO_4 \cdot NO_3^-$ clusters (Fig. 7k). Interestingly, diffusion mixing of $HNO_3$ into the centre axis is predicted to be minimal, centre line concentrations are modelled to be less than 1 permil of the sheath gas concentration (Fig. 7k). Likewise, the humidity of a moist sample flow is reduced by a dry sheath flow only marginally (Fig. 7e).

Figure 9 shows from both measurements and model predictions that significant transport of ions to the IMR and pinhole requires coordinated voltages $U_1$ and $U_2$ on the order of $-100\,\mathrm{V}$. For lower voltages, the electric transport of ions into the centre is too slow. For higher voltages, the ions are lost to the electrodes. Figure 9 further shows that voltage differences $U_2 - U_1$ of only a few V matter, and that the ion cage needs to be slightly more repulsive than the IMR cylinder. If the repulsion from the ion cage is too high (relative to the IMR cylinder), the ions are lost to the IMR cylinder. Vice versa, if the repulsive voltage of the ion cage is too low, the ions are lost to the ion cage. The model reproduces the measured general trend. The measured band width is slightly larger than model-predicted, likely because of small eddies within the inlet leading to transport additional to the idealised laminar flow. The subtly different slope is likely due to the exact insertion depth of the sampling tube into the assembly, which is known to affect the transmission.

The average reaction time $t_{\mathrm{avg}}$ for the standard setup is modelled to be $113\,\mathrm{ms}$ (eq. 7, Fig. 6), compatible with literature values of $160\,\mathrm{ms}$ (He et al., 2023). Interestingly, the average reaction time is about $10\,\mathrm{ms}$ longer than the geometrical time, derived from the centre flow velocity and length of the IMR. This extra time arises from the transport towards the centre line and the slower electro-convective transport at the entrance of the IMR. There is only negligible sensitivity of the reaction time to the magnitude of the chosen voltages.

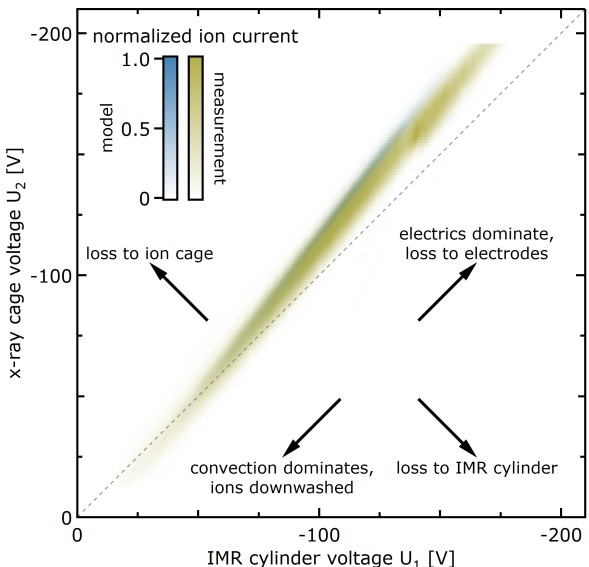

**Figure 9.** Sensitivity of pinhole $NO_3^-$ current towards IMR cylinder voltage $U_1$ and ion cage voltage $U_2$ in Eisele type inlet. Ion transport to the IMR and pinhole is only significant for a narrow combination of voltages that cause substantial electrophoretic transport while minimising surface losses.

## 4 Discussion

Table 1 lists similarities and differences between the MION2 and Eisele inlets. They both typically employ x-ray lamps as ionisation source, and use electric fields to transport reagent ions into the IMR. In the Eisele inlet, the ionisation volume is essentially electric field free; reagent ions and their complements are transported out of the ionisation volume by convection only. In the MION2 inlet, electric fields throughout the ionisation volume already separate ions of opposite charge, suppressing ion–ion recombination. The extracted ion concentration $c_S$ in the MION2 inlet is not higher than in the Eisele type inlet, as ion–ion recombination is not yet substantially cancelling the primary ion production rate of $6 \cdot 10^7 \, \mathrm{cm}^{-3} \, \mathrm{s}^{-1}$. The transfer from electric to advective ion transport is well-defined in the MION2 inlet as electric and advective streamlines are perpendicular to each other, whereas in the Eisele inlet an electro-convective streamline from the ion source volume to the centre line does not even exist due to the rotational symmetry (Fig. 7d). Accordingly, MION2 accomplishes a near-ideal ion delivery efficiency $\eta_D$ whereas Eisele reaches best ion delivery efficiencies $\eta_D$ of few $10\,\%$. Additional mixing (e.g., by eddies) in Eisele would not substantially increase the delivery to the pinhole, but could even dilute ion concentrations in the IMR. The combined differences in ion extraction and delivery lead to the pinhole ion concentration $c_P$ in MION2 to be approximately $40\,\%$ larger than in Eisele.

The reaction time in the Eisele inlet is 5 times longer than in the studied version of the MION2 inlet. This is a simple result of the geometry. At an axial advective velocity of $1.5 \, \mathrm{ms}^{-1}$, $15 \, \mathrm{cm}$ extra tubing correspond to a a lengthening of the reaction time by $100 \, \mathrm{ms}$. MION2 is routinely used with such drift tubes (He et al., 2023). The modelled reaction times and

**Table 1.** Characteristics and comparison of MION2 and Eisele inlet.

| metric | | Eisele | MION2 |
|---|---|---|---|
| creation of ions | | x-ray-irradiation of precursor gas | x-ray-irradiation of precursor gas |
| charge separation | | in peripheries of ionization volume | throughout ionization volume |
| extracted ion concentration[a] | $c_S$ | $6 \cdot 10^6 \, \mathrm{cm}^{-3}$ | $2 \cdot 10^6 \, \mathrm{cm}^{-3}$ |
| electro-convective coupling / ion injection | | counter-convective | perpendicular |
| IMR length / diameter | | $152 \, \mathrm{mm}$ / $44 \, \mathrm{mm}$ | $> 33 \, \mathrm{mm}$ / $22 \, \mathrm{mm}$ |
| ion delivery efficiency | $\eta_D$ | $\sim 10 - 20\%$ | $> 90\%$ |
| ion concentration at pinhole | $c_P$ | $1.1 \cdot 10^6 \, \mathrm{cm}^{-3}$ | $1.5 \cdot 10^6 \, \mathrm{cm}^{-3}$ |
| pinhole ion current | $I_{R\pm}$ | $1.5 \cdot 10^7 \, \mathrm{s}^{-1}$ | $1.7 \cdot 10^7 \, \mathrm{s}^{-1}$ |
| flow rate sample/sheath/total | | 10/20/30 slpm | 20/0/20 slpm |
| average reaction time | $t_\mathrm{avg}$ | $113 \, \mathrm{ms}$ | $\geq 22 \, \mathrm{ms}$ |
| detection limit $H_2SO_4$ (with $NO_3^-$) | $\Lambda$ | $7.6 \cdot 10^4 \, \mathrm{cm}^{-3}$ [b] | $6.1 \cdot 10^4 \, \mathrm{cm}^{-3}$ [b] |
| transport to IMR | | ions and reagent gas | ions only |
| rapid reagent switching | | incompatible | compatible |

a: using a primary production of $6 \cdot 10^7 \, \mathrm{cm}^{-3} \, \mathrm{s}^{-1}$

b: from He et al. (2023), scaling reported detection limit for Eisele inlet to MION2 inlet using relative IMR ion concentrations

ion concentrations are in line with literature comparing both inlets regarding their detection limits, finding that MION2 inlets enable lower detection limits if a similar reaction time is used (He et al., 2023).

Both Eisele type and MION2 inlets can in principle be used with different reagent ions, but by design only a single Eisele ion source of given reagent ion can be coupled to a mass spectrometer at a time. In contrast, multiple MION2 inlets can be coupled to a mass spectrometer at the same time, either of different reagent ion (Rissanen et al., 2019) or of same reagent ion but different reaction time (He et al., 2023). Rapid switching between ion sources is achieved by controlling the electric fields, i.e., grounding the deflector electrode effectively passivates the source. Field measurements with the Eisele type inlet are arguably not practical with reagent ions other than $NO_3^-$, requiring large quantities of ultra-pure sheath gas, rather than just filtered air.

Overall, the MION2 inlet is more efficient in charge separation, ion extraction and delivery, avoids the contamination of sample gas with reagent gas, enables ion switching, and allows for the adjustment of reaction time down to as little as $24 \, \mathrm{ms}$ (Rissanen et al. (2019), e.g. via insertion of KF25 drift tubes). MION2 is arguably more complex, as controlling the auxiliary flows in the ionisation part (reagent supply, purge supply, purge exhaust) is required, but it does not require a large sheath flow.

The model closure justifies the assumptions and simplifications made in the model. (1) Laminar gas flows are sufficient in the model to explain observations (Fig. 9), small scale eddies may be present but do not critically influence the operation. (2) The ionisation of the precursor gas is assumed to lead to primary production of ions that is constant throughout the ionisation volume (Fig. 1A); both the geometry of the irradiated volume and the constant rate are a simplification (Anttalainen et al.,

2021). The ion creation by photo electrons from the irradiation of the electrodes with x-ray is not considered. For the Eisele
inlet, only one x-ray source was used in the laboratory measurements. (3) The electric mobility is assumed to be identical for
all studied compounds; the mobility of the reagent ion directly affects what deflector voltage is required to achieve a proper
extraction from the ionisation volume and injection into the IMR; the mobility of other compounds matters as they influence
ion–ion recombination and secondary chemistry. While the model could still be refined regarding the above points, especially
the representation of turbulent flow, it already elucidates the limiting process in the current designs and shows potential for the
improvement in new inlet designs.

The model suggests that the source concentration $c_S$ is limiting the ion concentration in the IMR. Ion–ion recombination
or space charge are not yet significantly at play under the studied conditions with ion concentrations of few $10^6\,\mathrm{cm}^{-3}$, but
cannot be disregarded for moderately enhanced ion concentrations exceeding $\sim 10^7\,\mathrm{cm}^{-3}$. For the Eisele type inlet, the poor
ion delivery efficiency further reduces the attainable concentrations. Increasing ion concentrations in the IMR in new inlet
designs would require to enhance the source concentration (either via a larger primary production or extraction from a weaker
electro-convective field), while not compromising the efficient delivery. At high ion concentrations that lead to space-charge
induced electric fields approaching or exceeding the electrode-prescribed electric field (an ion concentration of $10^7\,\mathrm{cm}^{-3}$ is
equivalent to an electric field change of $18\,\mathrm{V}\,\mathrm{cm}^{-1}\,\mathrm{cm}^{-1}$), space charge ceases to be a minor perturbation to ion trajectories
but becomes their primary driver. Avenues for improvement (e.g., Ewing et al., 2023) will constitute a future study.

## 5   Conclusions

This study elucidates the inlet-internal processes, explains observed sensitivities, and highlights the design differences between
the MION2 and Eisele type inlet. While the Eisele type inlet performs well for a relatively simple setup, MION2 type inlets
extract ions more efficiently because of the electric field within the ion volume, and are near-ideal in delivering the ions to the
IMR. MION2 type inlets also allow ion-switching or the sampling of ambient ions. The finding that the ion delivery of MION2
is already near-ideal is curious and suggests that higher initial production rates are initially needed to substantially enhance the
reagent ion concentrations in the IMR. It is clear that the model will prove useful in the development of new inlet designs that
deliver ions at higher concentration or are simpler and more robust.

*Code and data availability.*   The data underlying the figures and information on the model will be provided on reasonable request.

*Author contributions.*   A.S., J.K. and M.S. conceived the study idea. H.F. performed the modelling and MION2 measurements and interpreted
the results with help from all co-authors. C.R. and N.S. acquired data with the Eisele type inlet. H.F. wrote the manuscript with help from all
co-authors.

*Competing interests.* Karsa Ltd. and Aerodyne Research Inc. are involved in the production and distribution of the Eisele and MION2 type inlets.

*Acknowledgements.* We thank Osmo Anttalainenen for initial help with COMSOL simulations. This work received support by the Finnish Research Impact Foundation and Research council of Finland (346370, 346373, 356134). This project has received funding from the European Research Council under the European Union's Horizon 2020 research and innovation programme under Grant No. 101002728.

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
