# Peer review of "Multiphysical description of atmospheric pressure interface chemical ionisation in MION2 and Eisele type inlets"

_Atmospheric Measurement Techniques, 2024_

## Author Comment (AC1)

**Quick reply to reviewer comment #2**

We appreciate the critical review. We deem it appropriate to provide a swift comment to clarify that there is no inconsistency with the fundamental model setup in the study. We are under the impression that the confusion caused to the reviewer stems from assuming certain approximations of the advective velocity profile in conjunction with the peak advective velocity, resulting in flow rates putatively incompatible with values from Table 1.

The modeling indeed uses the flow rates from Table 1 (20 slpm for MION, 30 slpm total for Eisele), consistent with what is in the literature.

The reviewer seems to use a constant velocity profile ("plug flow") to derive the high flow rates calculated in the comment (45 slpm for MION, 160 slpm for Eisele) that are indeed not compatible with the literature (too high). Assuming a parabolic velocity profile (a fully developed laminar flow) leads to flow rates that are lower by a factor of 2 (22 slpm for MION, 80 slpm for Eisele). For the MION inlet, this flow rate is compatible with the values shown in Table 1. For the Eisele inlet, this still seems incompatible. Here, it turns out that the velocity profile is neither constant nor parabolic, but rather behaves as depicted in Figure C1 (blue line). The flow velocity has a pronounced peak in the center, the velocity of the sheath flow is modeled to be rather weakly varying. This profile establishes as there is not much time for a fully parabolic flow to develop in the IMR after the core and sheath flow (green lines in Fig. C1) merge. Contributing to this non-parabolic profile is the fact that the velocity profile downstream of the IMR (close to the pinhole plate) is not parabolic, either. This is an interesting observation which is not obvious from the manuscript figures in their current form.

[Figure]

Figure C1: Flow velocity as function of radius (arc length) in the Eisele-type inlet. Profile before merging of sheath and core flow (green) and in center of IMR (blue). While the individual flow profiles are parabolic before merging, a parabolic flow in the IMR does not fully develop.

Effectively, we are confident that there is no fundamental problem with the study and that we will be able to clarify the profile of flows throughout the Eisele and MION inlet in a revised version of the manuscript that avoids confusion or misinterpretation. In due time, we will provide such a revised manuscript that also incorporates modifications motivated from the other reviewer comments.

---

## Author Comment (AC2)

**Final author comments**

We would like to express our gratitude to the reviewer for taking the time and providing comments and a critical review to the manuscript.

Below, we provide point-to-point responses to the comments in red and document changes to the manuscript in blue.

Additional to the changes in direct context of the reviews, we have corrected multiple typographical errors in the manuscript, harmonized the panel labels of Figures 1 and 7, corrected the affiliations for one co-author, and amended the acknowledgments.

Henning Finkenzeller, on behalf of all co-authors

**Reviewer #1 comment:**

**Report to the manuscript "Multiphysical description of atmospheric pressure interface chemical ionisation in MION2 and Eisele type inlets"**

This manuscript presents multiphysics simulations of two atmospheric pressure chemical ionization sources frequently used in analyzing atmospheric samples with a mass spectrometer. The authors validate the theoretical results with current measurements at specified electrodes and also compare them with general observations having made while working with these sources. In general, this manuscript provides interesting and valuable insights into the physical and chemical working principles of these devices. Hands-on visualizations of parameters impacting the operation are given.

In accordance with AMTs referee guideline, following aspects are addressed:

1. *Does the paper address relevant scientific questions within the scope of AMT?*

--- To users of these sources the presented simulations might be relevant, also in view of further improvements.

1. *Does the paper present novel concepts, ideas, tools, or data?*

--- According to the authors it is the first time that a multiphysics simulation tool has been applied to describe and visualize the physical and chemical principles of these devices.

1. *Are substantial conclusions reached?*

--- In my opinion, no further substantial conclusions are reached despite the insightful visualizations.

1. *Are the scientific methods and assumptions valid and clearly outlined?*

--- In principal yes, specific points are discussed further below.

1. *Are the results sufficient to support the interpretations and conclusions?*

--- In principal yes, specific points are addressed further below.

1. *Is the description of experiments and calculations sufficiently complete and precise to allow their reproduction by fellow scientists (traceability of results)?*

--- Yes.

1. *Do the authors give proper credit to related work and clearly indicate their own new/original contribution?*

--- Yes.

1. *Does the title clearly reflect the contents of the paper?*

--- Yes.

1. *Does the abstract provide a concise and complete summary?*

--- Yes.

1. *Is the overall presentation well-structured and clear?*

--- Appropriate.

1. *Is the language fluent and precise?*

--- Appropriate, specific points are addressed further below.

1. *Are mathematical formulae, symbols, abbreviations, and units correctly defined and used?*

--- Yes.

1. *Should any parts of the paper (text, formulae, figures, tables) be clarified, reduced, combined, or eliminated?*

--- Generally okay, specific clarification is addressed further below.

1. *Are the number and quality of references appropriate?*

--- Yes.

1. *Is the amount and quality of supplementary material appropriate?*

--- Not applicable.

**In compliance with the AMT referee guideline I do recommend publishing this article, however, with some minor changes/additions and requested comments presented in the following:**

**(i)** lines 147 and 148: Please comment on the measured ion currents in the order of $10^{-11}$ A. The used Tenma (72-2595) can only measure in the µA range according to its specification.

The reviewer is correct in that the regular measurement range of the multimeter in current mode only extends down to µA. Here, we measured in voltage mode the voltage-drop over the multimeter-internal 10 MOhm resistor. Usually, the 10 MOhm present an "infinitely large resistance" for measuring voltages, and the voltage drop across the resistor is essentially the source voltage which is meant to be measured. Here, adding the resistor does not introduce a voltage-drop significant enough to bias the system (burden voltage of single mV out of hundreds or few thousands V) but the voltage-drop is measurable.

We agree that this could be made clearer in the manuscript and revise the passage as follows:

The currents to the two topmost electrodes of the MION2 inlet ($4 \cdot 10^{-11}$ A, attraction of $H^+$) and for the ion cage of the Eisele inlet ($6 \cdot 10^{-11}$ A, attraction of $H^+$, negligible

adsorption of $NO_3^-$ ) were determined via the voltage-drop across the internal 10. MΩ resistor dedicated to measure voltages in a simple multimeter (Tenma 72-2595). The voltage-drop of 0.6 mV is measurable by the voltmeter and does not constitute a measurement bias under the test conditions.

(ii)    line 60: "…by 40…" please revise

Added missing "%."

(iii)    line 76: "a narrow…(A)…" please revise

Revised description to clarify analogy as follows:

Analogous to narrowing riverbanks that increase the water flow velocity (v) by reducing the cross-section area of the flow (A) but do not change the composition of the water (c), electric fields defined by electrodes affect the ion trajectories without changing their concentration.

(iv)    fig 4d and e: $U_A$ = -3000 V instead of $U_A$ = 3000 V

Added missing minus signs.

(v)    line 202: "The pinhole current is measured at a higher voltage than predicted." Please clarify.

We modified and expanded the discussions of Figure 5 substantially by a paragraph as follows to improve clarity:

The maximum pinhole current is measured at a higher accelerator voltage than predicted. Although the deflector voltage $U_D$ was chosen in the measurement to maximise the ion delivery, it is possible that the ion beam was not always axially centred to be contained within or to fully illuminate the pinhole flow. In very narrow ion beams (low $U_A$, compare Fig 4a) that are not aligned with the flow going to the pinhole, most ions entering the IMR would be lost to the exhaust flow ($J_E$). This is a plausible explanation for the gradual onset of the observed measured total pinhole current. We note that the measured pinhole current in Figure 5 apparently decreases faster towards higher accelerator voltages $U_A$ than expected from model simulations. Insufficient centring of the beam does not explain this observation, as the ion concentration within the beam varies only slightly. While shying from pinpointing a specific mechanism, we hypothesise that the effect originates in the volume controlled by $U_A$, i.e., the ionisation volume or the buffer volume between the ionisation volume and IMR. Actual ion mobilities considerably higher than assumed in the model at high $U_A$ would lead to a more rapid decrease. A lower degree of reagent ion hydration and cluster formation between the reagent ion and reagent gas at high field strength would increase the effective electrical mobility. The field strength sensitivity of the reagent ion mobility itself is less likely to be significant because of the still relatively weak field strength (Viehland and Mason, 1995). Space charge losses during transport (especially within the IMR) are found to be not yet significant for the ion concentrations of few $10^6$ cm$^{-3}$. In the model, the ion delivery efficiency $\eta_D$ (eq. 4) is larger than 90 %, essentially unity, for $|U_A|$ > 3000 V: The ion concentration is maintained from the ion source to the pinhole.

(vi)    line 204: "This could lead to a softening of the voltage sensitivity." Please clarify.

See comment v.

(vii)    Please clarify the argumentation for the significantly differing slope of the measured pinhole current in contrast to the simulated ion concentration in figure 5.

See comment v.

(viii)    line 288: "cm$^{-3}$" instead of "cm$^{-1}$"

Changed.

---

## Author Comment (AC3)

**Final author comments**

We would like to express our gratitude to the reviewer for taking the time and providing comments and a critical review to the manuscript.

Below, we provide point-to-point responses to the comments in red and document changes to the manuscript in blue.

Additional to the changes in direct context of the reviews, we have corrected multiple typographical errors in the manuscript, harmonized the panel labels of Figures 1 and 7, corrected the affiliations for one co-author, and amended the acknowledgments.

Henning Finkenzeller, on behalf of all co-authors

**Reviewer #2 comment:**

This paper compares two atmospheric pressure chemical ionization geometries using a detailed model simulations. Results from ion current measurements are presented and compared to simulations of the two different geometries. The overall goal of this manuscript is interesting and the results would be useful to the community. However I find the paper inconsistent or confusing at the least.

From my understanding, the advective velocities used in the model (Figures 1 and 7) are too large for the typical operation of either case of the CIMS. This is especially true for the Eisele design. For the MION described in Wang et al. [2021] with a 22 mm flow tube, a velocity of 2 m s$^{-1}$ equates to a total flow of ~45 slpm which is on the higher end of the 20-30 slpm typically described (Wang et al. [2021] used 32 slpm for example). For the Eisele design using the 44 mm diameter flow tube described in the model, 1.7 m s$^{-1}$ equates to a total flow of ~160 slpm. Well above the 20-45 slpm typically described (Tanner and Eisele [1995] and Sipila et al. [2018] used 45 and 30 slpm respectively). Later however, in Table 1, values more consistent with previously published operating parameters are presented for both systems.

The authors state that a reaction time of 113 ms is assumed for the Eisele configuration. Using a flow of 30 slpm as stated in Table 1 and the 44 mm diameter, a flow velocity of ~30 cm s$^{-1}$ is obtained, yielding a reaction distance of only 3-4 cm? The physical length of the actual IMR cylinder is ~15 cm. The 113 ms reaction time IS greater than the calculated ~88 ms geometric reaction.

This inconsistency/confusion furthered in the topic of turbulence. The authors state that a Reynolds number, $R_e$, of ~1600 was assumed. Using 45 slpm in a 22 mm dia. flow tube, as in the MION model, yields a $R_e$ of ~2800, well into the transition zone towards turbulent flow. The 160 slpm flow in a 44 mm flow tube, as in the Eisele model, yields a $R_e$ >4000, well into the turbulent flow regime. If, however, the flows from Table 1 are used laminar conditions are maintained.

Overall, as a person familiar with both systems, I find this work inconsistent and/or confusing and hard to follow. The manuscript should be rewritten with an eye towards addressing these inconsistencies or making things more understandable. The work is interesting and will provide insight for users of either technique. The visualizations are quite useful. However, it needs more work before I would recommend publication.

Following the reasoning outlined in the "Quick reply to reviewer comment #2", we have revised the manuscript to clarify the flow rates used in the modelling, the flow profiles, and that using laminar flow in the model is justified. Specifically, we addressed these issues in the following sections:

1. Clarification of flow rates for each inlet, Section 2.2 (Model setup)

   For the modelling of the Eisele-type inlet, 10 slpm sample, 20 slpm sheath, and 1 slpm flow to the mass spectrometer are used (Tanner and Eisele, 1995). For the MION2 inlet, 20 slpm exhaust flow (Wang et al., 2021) and 0.8 slpm flow to the mass spectrometer are used. The auxiliary reagent, purge, and reagent exhaust flow are $J_R$ = 10 smlpm, $J_{RE}$ = 50 smlpm, and $J_{RP}$ = 100 smlpm.

2. Clarification of flow profiles in both MION2 and Eisele-type, and respective Reynolds numbers (including the parameters used for their derivation), Section 3.1 and 3.2 (Results)

   MION2, Section 3.1 (Results MION2 inlet):

   Assuming an interface upstream of the MION2 inlet that creates laminar flow (Reynolds number Re ≈ 2100 (using D = 20 mm, u = 1.6 m s$^{-1}$, v = 1.48 · 10$^{-5}$ m$^2$ s), the flow velocity profile is parabolic throughout the sample tube and IMR close up to the pinhole plate, where the flow splits to the exhaust and pinhole.

   Eisele-type, Section 3.2 (Results Eisele type inlet):

   Considering the paramount importance of the flow field throughout the inlet for the gas and ion transport, we added Figure 8 and an accompanying paragraph, which effectively communicates the modelled velocity profile within the Eisele-type inlet before and after merging the sample and the sheath flow.

[Figure]

Fig. 8: Modelled velocity profile within the Eisele-type inlet before and after merging sample and sheath flow, using 10 slpm sample flow and 20 slpm sheath flow. The composite profile establishing in the IMR has a pronounced maximum in the centre and a rather flat shoulder.

Figure 8 shows the velocity profile of the advective velocity upstream of the flow merging at the x-ray lamp plane and downstream after mixing at the IMR mid plane. Before merging, the flow profiles within the different channels are near-parabolic, as expected for fully developed laminar flows. After merging, the individual flow profiles combine to form a transition composite that maintains a near-parabolic shape in the innermost 5 mm with a pronounced maximum at the centre line and a rather flat shoulder with low velocity. The profile is the result of the relatively little interaction with the IMR surface after merging: The IMR radius (22 mm) is relatively large in comparison to the total IMR length (15 cm). Additionally, the velocity profile at the downstream end of the IMR (close to the pinhole plate) is not parabolic, either. The Reynolds number Re $\approx$ 1840 (using D = 16 mm, u = 1.7 m s$^{-1}$, v = 1.48 $\cdot$ 10$^{-5}$ m$^2$ s) at the downstream end of the sample tube, the location most prone to cause turbulence, supports assuming laminar flow in the modelling.

We remain convinced that there is no fundamental problem with the study and believe to have appropriately addressed the raised concerns about the clarity of presentation and consistency of the study.

---

## Author Response (AR2)

*Public justification (visible to the public if the article is accepted and published):*
*The reviewer comments and criticisms have been adequately answered in the revised manuscript, however some specific points concerning fluid dynamical details etc. should be further refined in the final manuscript.*

*Additional private note (visible to authors and reviewers only):*
*Dear Authors,*
*while I think the reviewer comments and criticisms have been adequately answered in the revised manuscript I need to ask you to clarify some additional points that are not clearly explained in my view.*

We thank the editor for diligently reviewing the manuscript and providing detailed comments. We hope to adequately address the remaining concerns with the point-by-point replies below and modifications to the manuscript.

*1. Line 127 states that the model assumes laminar flow but for my understanding this is not given at the involved flows and geometries. The entrance length for either IMR is in excess of 1m, so a laminar profile will not at all be developed at the end of the concentric flow part of the Eisele-IMR nor at the orifice of both IMRs. The effect of the flow profiles for the resulting profile shown in Fig.8 and the final results should be discussed.*
We agree that the entrance length for both IMRs exceeds 1 m (under the assumption of laminar flow) and agree with the conclusion that there is often not enough time in the different inlet sections for the laminar flow to completely develop into a fully developed laminar flow profile, including the volumes close to the orifices. As we understand the comment, the editor's principal concern seems to be whether the flow profiles at the merging of the sheath and sample flow (new Fig. 8) are reasonable.
The velocity profile of the sample flow entering the sample tube is model-initialised as fully developed. This is justified if an appropriate sampling tube extending the inlet is used in actual measurements. We amend the description of the model setup accordingly.
The initial flow profile of the sheath flow (at the very left of the geometry shown in Fig. 7) is likewise model-initialised as fully developed. In reality, the profile - resulting from the provision of the sheath gas through tubing, a hole-filled plate ("shower head"), and finally a laminarising mesh – is somewhere between a plug and parabolic flow profile. While the assumption of a fully developed flow is arguably less accurate for this very initial section of the sheath flow, we deem that this does not limit the overall accuracy of the modelling further downstream. Figure 8 shows how the initially parabolical sheath flow profile (for the entire radial range 9–22 mm) splits into two fairly parabolic profiles (9–15 mm and 16–22 mm, divided by the "ion cage electrode") over the distance of several cm only. The reason for the relatively fast development of the profiles is the close distance (5 mm) between the surfaces, equivalent to an entrance length of many cm only. The model does not assume a fully developed laminar flow in this section (or elsewhere), the resulting flow profiles are merely the result of the modelled shear forces under the assumption of laminar flow. We conclude that assuming an initially fully developed laminar sheath flow does not present a limitation, and do not think there is a fundamental issue with the study.

Multiple adjustments that aim for more clarity in the text include:
Section 3.1, MION2 inlet, line 166 in change-tracked manuscript:
Assuming an interface upstream of the MION2 inlet that creates a fully developed laminar flow, …
Section 3.2 Eisele type inlet, line 224 in change-tracked manuscript:

The initial sample flow velocity profile is assumed to be fully developed, assuming an appropriate interface upstream of the inlet. The sheath flow profile, initialised likewise as fully developed laminar flow, quickly adjusts to the concentric tubing geometry.

*2. Since it is far from easy to quantitatively generate dilute H2SO4 mixtures the procedure used should be at least given and referenced. The consequences for the reported sensitivities should be discussed as well.*

We would like to clarify that $H_2SO_4$ was not used in any experiments within this study, only in the modelling. In the modelling, $H_2SO_4$ is used as a prototypical molecule that clusters with the reagent ion. The abundance of the $H_2SO_4$ in the sample gas does not matter and will only proportionally affect the abundance of the formed cluster, as long as there is no substantial reagent ion depletion. We intend to make this clearer with a few minor changes in the manuscript.

(1) Section 2.2 now elaborates on why the dilute concentration was chosen.

As proxy for target molecules, dilute sulfuric acid $H_2SO_4$ is modelled to be contained in the sample flow at a mixing ratio of 1 ppt. It reacts kinetically with $Br^-$ and $NO_3^-$ to form $H_2SO_4.Br^-$ and $H_2SO_4.NO_3^-$. The magnitude of the $H_2SO_4$ abundance is not critical for the interpretation of the modelling results, as long as the clustering with the reagent ion does not substantially reduce the reagent ion concentration.

(2) Section 2.3 Laboratory measurements:

$H_2SO_4$ or other targets gases were not employed in the laboratory experiments but treated in the modelling only.

(3) We have amended the captions of Fig. 1 and Fig. 7 to now indicate that the figures show modelled physical quantities, not measured physical quantities.

*Minor issues:*

*1. For the MION2 length should be also given at the beginning of section 3.1, now it is only introduced somewhere later in the text. The basic geometry data of both IMRs could also be included in Table 1 for convenience. Fig. 1 itself does not give any details (line 160).*

We now specify the length of the MION2 and Eisele IMR, in addition to the IMR diameter, in the respective sections 3.1 and 3.2. We are happy to follow the suggestion to add the information in Table 1.

*2. I would find it interesting to see the downstream circular cross sections of the product ions few cm in front of the orifices within Figs. 1 and 7.*

The model framework allows to extract this information. The reagent ions distribution is the following.

[Figure]

Figure R1: Reagent ion concentrations ($NO_3^-$, [$cm^{-3}$]) 5 mm in front of the orifice plate, for MION2 (a) and Eisele-type inlets (b). The axes indicate the radial distance [mm]. The distribution is essentially rotationally symmetric for both inlets. In MION2, the ion beam is marginally compressed in the y-direction (the direction of ion injection into the IMR).

The distribution of cluster ions is of similar to that of the reagent ions, given that the reaction time between the target gas and the reagent ion is approximately path independent up close to the orifice. We deem the information that Fig. R1 could add to the manuscript too little to warrant the inclusion of such a dedicated figure in the manuscript. Likewise, we are reluctant to add the cross sections as additional panels to figures 1 and 7, as it would make the figures even more busy than what they already are. However, based on the editor suggestion, we consider it prudent to include the cross sections in Fig. 4 – in which the change of the cross-section area of the ion plume is discussed – and briefly discuss the rotational symmetry in the main text. The base value of $U_A$ in Fig. 4 was updated to -1500 V (previously -3000 V), to be directly comparable to Fig. 1.

Line 199 of the change-tracked manuscript:
If chosen correctly, the electro-advective streamlines connect the pinhole and the ionisation volume (Figure 1d), and the distribution of ions in the IMR close up to the pinhole is essentially rotationally symmetric. The marginal beam compression in the ion injection direction is due to the advective velocity being largest in the plane of injection.

[Figure]

Figure 4: Sensitivities of $NO_3^-$ concentrations in MION2 inlet to different acceleration voltages $U_A$ (a-c), deflector voltage $U_D$=0 V for deactivation (d), and primary ion production rate (e). The semi circle areas show the ion concentration in the cut plane 5 mm in front of the orifice. The colour scale ranges from 0 to the maximum described in each panel. Figures a-d use the same colour scale. The width of the ion beam increases for larger voltages, while the extracted concentrations slightly decrease. At concentrations of $10^7$ $cm^{-3}$ space charge leads to a spreading of the ion beam, the concentration at the pinhole is lower than at the ionisation volume.